# Open-Ended Learning in General-Sum Games: The Role of Diversity in Correlated Equilibrium

## Abstract

The primary in this work focuses on the challenging and crucial task of identifying and selecting equilibria for $n$-player general-sum games. PSRO serves as a comprehensive framework for tackling complex games by leveraging the concept of the meta-game. However, prior research on PSRO mainly concentrates on solving two-player zero-sum games. Extended approaches such as JPRSO and $\alpha$-Rank can address multi-player general-sum games, and these methods theoretically ensure uniqueness and convergence. Nonetheless, a noticeable gap often exists between the joint policy distribution derived by the solver and the target equilibrium, which can undermine the robustness of the joint policy. Within the PSRO framework, diversity characterizes the distinctions among policies within the population, representing the exploration of the policy space by players. Consequently, allocating greater sampling probabilities (meta-strategy) to more diverse policies encourages players to employ more exploratory policies, thereby mitigating the risk of exploitation. We begin by incorporating diversity measures into solving equilibria for $n$-player meta-games and introduce a novel equilibrium concept, called Diverse (C)CE, the objective of which is to maximize sum of expectations of each player's diversity. In alignment with this, we present a policy training algorithm, Diverse Correlated Oracle (DCO), which effectively associates policy dynamics with the joint policy distribution. The experimental results conducted on a range of multi-player, general-sum games demonstrate that our algorithm outperforms JPSRO and $\alpha$-Rank and enhances the approximation of the joint policy distribution towards the target equilibrium by notably reducing the gap.

## 1 Introduction

Solving equilibria for multi-player general-sum games has consistently been a matter of broad interest. Nevertheless, advancements in solving two-player zero-sum games have outpaced those in addressing $n$-player general-sum games, primarily due to constraints related to problem size and computational complexity. In two-player games, the common objective is typically to identify a Nash Equilibrium (NE) (Nash, 1951), a problem that proves to be relatively straightforward in transitive games (Perez-Nieves et al., 2021). Consequently, recent research has placed a greater emphasis on games featuring non-transitive components, where winning policies exhibit noticeable circularity(Balduzzi et al., 2018a). An outstanding frameowrk is Policy Space Response Oracle (PSRO) (Lanctot et al., 2017), which typically achieves improved approximations of the Nash equilibrium (NE) by promoting diversity (Balduzzi et al., 2019; Perez-Nieves et al., 2021; Liu et al., 2021; 2022; Yao et al., 2023) within the policy population during the Best Response (BR) solving process.

Extensions of PSRO, such as JPSRO and $\alpha$-Rank (Muller et al., 2019; Omidshafiei et al., 2019), have advanced the solution of equilibria in $n$-player general-sum games. Analyzing and operationalizing the task of finding an equilibrium in $n$-player general-sum games is inherently difficult (Harsanyi et al., 1988; Daskalakis et al., 2009; Avis et al., 2010; Goldberg et al., 2013). In most cases, such games typically encompass multiple equilibria, and a crucial step is to select among these equilibria. Moreover, the computational expense of finding a NE in $n$-player general-sum games becomes increasingly prohibitive. To address this issue, Joint PSRO (JPSRO) extends PSRO from two-player games to $n$-player games, by utilizing (Coarse) Correlated Equilibrium ((C)CE) (Aumann, 1974;

Moulin & Vial, 1978) as the target equilibrium space. Additionally, inspired by evolutionary dynamics, $\alpha$-Rank functions as a descriptive method for delineating the ranking of policies in multi-agent interactions. It utilizes Markov Conley Chains to represent the dynamic process of the game and attempts to calculate a fixed distribution for the ranking. This approach facilitates an understanding of the underlying dynamics and convergence of these policies (Omidshafiei et al., 2019).

While these methods provide theoretical convergence guarantees, it doesn't necessarily ensure that the equilibrium can be precisely approximated in practical scenarios. Consistent with the Maximum Entropy Correlation Equilibrium (MECE) (Ortiz et al., 2007), JPSRO formulates a convex objective function on the joint policy, utilizing the Gini impurity (Marris et al., 2021; Breiman, 2017; Biship, 2007), to address the challenge of selecting a unique equilibrium within the game. Nonetheless, maximizing entropy or Gini impurity simply allows each policy in the population to be sampled with a more equal probability, even if some policies can't contribute to convergence to the equilibrium. In addition, as far as $\alpha$-Rank is concerned, its computational complexity is still very high (Yang et al., 2019), which makes the problem difficult to be solved in a limited time.

Given that diversity has the attributes of promoting exploration within the policy space and being computationally tractable, we leverage it to reduce gap and promote convergence when approximating equilibrium in multi-player general-sum games. Additional supplementary information that characterizes policy dynamics proves advantageous for enhancing the robustness and approximation of equilibrium solutions, particularly in the multi-player games. There are numerous methods describing policy dynamics within a game, including approaches like Graphical Models (Kearns et al., 2013; Garnelo et al., 2021) and Agent Modeling (Albrecht & Stone, 2018; Li & Miikkulainen, 2018). Nonetheless, these methods typically rely on a substantial amount of prior knowledge (Albrecht & Stone, 2018) and often entail high computational overhead. Diversity quantifies differences among policies in either the payoff space or the state-action space. When more diverse policies are assigned higher sampling probabilities, it encourages players to prefer these policies, promoting further exploration of the policy space and reducing the possibility to be exploited by other players (Wu et al., 2022). Furthermore, in comparison to alternative methods, it offers the advantage of being notably lightweight and computationally efficient.

In accordance with the aforementioned context, we pioneer the application of policy diversity to solve the equilibrium of meta-games, introducing a novel concept named Diverse (Coarse) Correlated Equilibrium (Diverse (C)CE). Subsequently, we innovate by formulating an adapted policy population training framework, Diverse Correlated Oracle (DCO), for the new equilibrium. Furthermore, in this paper, we substantiate the advantages of our approach through experimental results. Based on these insights, we make the following contributions:

- We extend the concept of diversity measures from 2-player games to $n$-player games. In $n$-player games, policy diversity can be conceptualized as a multivariate function, with each policy from the policy population serving as an independent variable, while considering the policies of other opponents as specified parameters.

- Leveraging policy diversity within $n$-player games, we introduce a novel equilibrium concept referred to as the Diverse (Coarse) Correlation Equilibrium (D(C)CE). This equilibrium offers a computationally more tractable alternative to Nash Equilibrium (NE) and proves suitable for addressing multi-player general-sum games.

- We integrate D(C)CE into solving equilibria for $n$-player meta-games, leading to the development of a new policy training algorithm known as Diverse Correlated Oracle (DCO). The algorithm introduces a novel perspective by effectively associating the dynamics of players' policies with the joint policy distribution.

## 2 NOTATIONS AND PRELIMINARIES

This section introduces correlated equilibrium, the training algorithm (J)PSRO for multi-player games, and existing policy diversity measures.

## 2.1 NOTATIONS OF (META-)GAMES

Games are modelled to describe interactions between multiple agents, which can be defined as a tuple $\langle \mathcal{N}, \mathcal{S}, P, \mathcal{A}, u \rangle$. $\mathcal{N}$ denotes the set of players. Players will take actions $a(s) = (a_1(s), a_2(s), ..., a_n(s)) = (a_i(s), a_{-i}(s)) \in \mathcal{A}(s) = \otimes_i \mathcal{A}_i(s)$ on a given state $s = (s_1, s_2, ..., s_n) \in \mathcal{S}$, where $s_i \in \mathcal{S}_i$. $P : \mathcal{S} \to \mathcal{N} \cup \{C\}$ is the player function which determines which player to take an action on $s$, where $C$ is the chance node. For multi-player games, we denote the policy profile as $\pi = (\pi_1, \pi_2, ..., \pi_n) = (\pi_i, \pi_{-i}) \in \Pi$, where $-i = \mathcal{N} \backslash \{i\}$. $u : \Pi \to \mathbb{R}$ is the utility function and the payoff of player $i$ when all players follow policy $\pi$ is $u_i(\pi)$.

The meta-game can be thought of as a "game of games", denoted by $\langle \mathcal{N}, \Pi, \mathcal{M} \rangle$. It can be regarded as a normal-form game, where an action $a$ is replaced by a policy $\pi$ and a meta-strategy is used to denoted the probability distribution over policies. Let $\sigma$ be a meta-strategy over $\Pi$, where $\sigma[k, l, ...q] = \mathbb{P}(\pi_i = \pi_i^k, \pi_j = \pi_j^l, ..., \pi_n = \pi_n^q)$. $\mathcal{M} = \otimes_i \mathcal{M}_i$ is the payoff tensor on the joint policy set $\Pi$, where $\mathcal{M}_i[k, l, ...q] = u_i(\pi_i^k, \pi_j^l, ..., \pi_n^q)$.

## 2.2 CORRELATED EQUILIBRIUM

A Correlated Equilibrium (CE) (Farina et al., 2020; Marris et al., 2022) refers to a joint (meta-)strategy $\sigma^*$ in which no player can unilaterally change $\pi_i$ to $\pi_i'$ to obtain higher returns. Actually, the approximate correlation equilibrium ($\epsilon$-CE) is more widely used. We denote the advantage function of player $i$ switching from $\pi_i$ to $\pi_i'$ as $A_i(\pi_i', \pi_i, \pi_{-i}) = u_i(\pi_i', \pi_{-i}) - u_i(\pi_i, \pi_{-i})$. Then for $\forall i \in \mathcal{N}, \pi_i' \neq \pi_i \in \Pi_i$, the $\epsilon$-CE can be described as:

$$\sum_{\pi_{-i}} \sigma(\pi_{-i}, \pi_i) A_i(\pi_i', \pi_i, \pi_{-i}) \leq \epsilon, \tag{1}$$

which can be written as $A_i \sigma \leq \epsilon$ for $\forall i$. The result of these linear inequality constraints is that the set of valid CE solutions is a convex polytope in linear space.

A special case is $\sigma(\pi) = \prod_i \sigma(\pi_i)$, which implies $\sigma$ is a Nash Equilibrium (NE). The NE is located on the boundary of polytope of the solution space, for non-trivial games (Nau et al., 2004). An important property is that the maximum sum of social welfare under CEs is greater than or equal to that under NEs.

Additionally, Coarse Correlation Equilibrium (CCE) (Moulin & Vial, 1978) is a looser equilibrium of which CE is a subset: NE $\subseteq$ CE $\subseteq$ CCE . But players may only consider deviating until they receive a recommended policy. Therefore the gain from deviating before a policy has been recommended is defined as $A_i(\pi_i', \pi) = u_i(\pi_i', \pi_{-i}) - u_i(\pi)$. Then the $\epsilon$-CCE can be described as:

$$\sum_{\pi} \sigma(\pi) A_i(\pi_i', \pi) \leq \epsilon. \tag{2}$$

## 2.3 (JOINT) POLICY-SPACE RESPONSE ORACLES

Policy space response oracle (PSRO) (Lanctot et al., 2017) is a population-based multi-agent policy learning method. PSRO is mainly used to solve players' policy on the meta-game iteratively. The algorithm initializes a policy pool $\Pi_i$ for every player. A new policy is added in each iteration. Specifically, PSRO first uses Meta-Solver (MS) to calculate the distribution of Best Response (BR) policies, followed by RL or accurate calculation to obtain the BR $\pi_i^{t+1} \in \arg\max_{\pi_i^* \in \Pi_i^*} \sum_{\pi_{-i} \in \Pi_{-i}^{0:t}} \sigma^t(\pi_{-i}) u_i^*(\pi_i^*, \pi_{-i})$. The procedure repeats until the meta-game NE converge.

Joint PSRO (JPSRO) (Marris et al., 2021) extends PSRO to multi-player games, where (C)CE, instead of NE, is solved by MS. And the BR operator is also made a corresponding correction for CE: $\pi_i^{t+1} \in \arg\max_{\pi_i^* \in \Pi_i^*} \sum_{\pi_{-i} \in \Pi_{-i}^{0:t}} \sigma^t(\pi_{-i}|\pi_i) u_i^*(\pi_i^*, \pi_{-i})$.

## 2.4 Existing Policy Diversity Measures for PSRO

In 2-player zero-sum games, the diversity of player $i$ is denoted by $f_{i|l} = f(\pi_i^1, \pi_i^2, ....)$, given the population $\Pi_i = \{\pi_i^1, \pi_i^2, ...\}$ and the opponent's policy $\pi_{-i}^l$. Existing diversity metrics can be mainly classified into Behavioral Diversity (BD) and Response Diversity (RD) (Liu et al., 2021). BD measures diversity by characterizing the difference in state-action distribution, such as Occupancy Measure Mismatching (Liu et al., 2021), Policy Space Diversity (Yao et al., 2023), etc. RD, including Effective Diversity (Balduzzi et al., 2019), Expected Cardinality(Perez-Nieves et al., 2021), Population Diversity (Parker-Holder et al., 2020), Convex Hull Enlargement (Liu et al., 2021), constructs the diversity over the empirical payoff matrix through different payoffs while encountering multiple opponent policies. Besides, Unified Diversity Measure (Liu et al., 2022) provides a unified view of BD and RD. Our work mainly involves Occupancy Measure Mismatching and Convex Hull Enlargement, and the combination of them is called **Combined Diversity Measure**.

**Convex Hull Enlargement** seeks to enlarge the convex hull of the policy represented by the payoff vector as much as possible. It is defined as a form of Euclidean projection:

$$\text{Div}(\Pi_i \cup \{\pi_i'\}) = \min_{\mathbf{1}^T \beta = 1, \beta \geq 0} ||\mathcal{M}_i^\top \beta - \mathbf{m}||, \tag{3}$$

where $\pi_i'$ is the new policy, and $\mathbf{m}$ is the payoff vector of $\pi_i'$ against each opponent policy in $\Pi_{-i}$: $\mathbf{m}[j] = u_i(\pi_i', \pi_{-i}^j)$.

**Occupancy Measure Mismatching** is to maximize the difference between the occupancy measure corresponding to the meta-strategy under NE and that corresponding to the new joint policy:

$$\text{Div}(\Pi_i \cup \{\pi_i'\}) = D_f(\rho_{(\pi_i', \pi_{-i}^*)} || \rho_{(\pi_i^*, \pi_{-i}^*)}), \tag{4}$$

where $D_f$ is a general $f$-divergence between the two distributions, $\pi_i'$ is the added policy. $\pi_i^*$ and $\pi_{-i}^*$ are average policies weighted by a meta-strategy NE $(\sigma_i^*, \sigma_{-i}^*)$ on $\mathcal{M}_i$.

## 3 Diverse Correlated Equilibrium for Meta-Games

Entropy and Gini impurity are frequently employed to quantify the uncertainty of events (Shannon, 1948; Biship, 2007; Breiman, 2017). The sampling probability of the policy is closer to the uniform distribution, the entropy or Gini impurity is larger. From this perspective, the purpose of both MECE and MGCE is to try to give each policy an equal opportunity to be sampled. These methods do not increase the probability of sampling more diverse policies, thus they do not contribute to the player's exploration of the policy space.

Most of studies about policy diversity focus on the solution of BR. By using diversity measure $f$ as a regular term, players are encouraged to explore more diverse policy populations to reduce the population exploitability (PE) and promote convergence to full game equilibrium (Yao et al., 2023). However, these works do not allocate larger sampling probabilities ($\sigma$) to more diverse policies, limiting the potential for these policies to play a more significant role in encouraging exploration within the policy space and improving equilibrium robustness.

We utilize policy diversity measures denoted as $f$ to resolve the equilibrium $\sigma^*$ of meta-games. Concretely, we introduce a new correlated equilibrium concept referred to as Diverse (Coarse) Correlated Equilibrium (D(C)CE), which is versatile enough to address not only two-player zero-sum games but also $n$-player general-sum games. The primary objective of this equilibrium is to maximize the sum of expectations of each player's policy diversity.

### 3.1 Forms of D(C)CE

We first extend the definition of diversity from two-player games to multi-player games. For player $i$'s policies $\Pi_i = \{\pi_i^1, \pi_i^2, ...\}$, its diversity measure is denoted by $f_{i|l,m,...q} = f(\pi_i^1, \pi_i^2, ....)$, given the opponents' policies $\Pi_{-i} = \{\pi_j^l, \pi_k^m, ..., \pi_n^q\}$.

The equilibrium of meta-games is built over the policy population of all players. Consequently, the isolation diversity term is meaningless for the solution of equilibrium. We are more concerned about the sum of each player's diversity expectations. The policy diversity expectation $f_i$ of player $i$ given opponent policies $\Pi_{-i}$ is denoted as:

$$
\begin{aligned}
f_i &= \sum_{l,m,\dots,q} \sigma_{-i}(\Pi_{-i} = \{\pi_j^l, \pi_k^m, \dots, \pi_n^q\}) f_{i|l,m,\dots q} \\
&= \sum_{l,m,\dots,q} \{\sum_k \sigma[k,l,\dots,q]\} f_{i|l,m,\dots q}.
\end{aligned}
\tag{5}
$$

Thus the sum of each player's diversity expectations is:

$$
\sum_i f_i = \sum_i \sum_{k,l,m,\dots,q} \sigma[k,l,\dots,q] f_{i|l,m,\dots q},
\tag{6}
$$

which can be formulated as $\mathbf{f}^\top \sigma$, where $\mathbf{f}$ and $\sigma$ should be interpreted as vectors of size $[|\Pi|]$.

Our objective is to increase the sampling probability assigned to diverse policies. This is achieved by maximizing the sum of each player's policy diversity expectations, thereby ensuring that the more exploratory policies play a more significant role in approximating the full game (C)CE. Inspired by JPSRO (Marris et al., 2021), we use a quadratic function as the optimization objective. The most intuitive form is $\max(\mathbf{f}^\top \sigma)^2 = \max \sigma^\top \mathbf{f}\mathbf{f}^\top \sigma$. However, this objective function does not have a unique solution. Therefore, it is reasonable to consider a combination of optimization objectives that can ensure the uniqueness of the solution. For this purpose, we combine the quadratic term $\sigma^\top \mathbf{f}\mathbf{f}^\top \sigma$ with the objective of MG(C)CE $-\sigma^\top \sigma$. Hence, the standard form of our method can be described as a programming problem comprising a quadratic objective function and linear constraints. We call the resulting solution concept Diverse (Coarse) Correlated Equilibrium (D(C)CE).

$$
\text{Quadratic objective: } \max -\frac{1}{2}\sigma^\top (I - \kappa \mathbf{f}\mathbf{f}^\top)\sigma,
\tag{7}
$$

$$
\text{(C)CE constraints: } A_i \sigma \le \epsilon \quad \forall i \in \{1,2,\dots,n\},
\tag{8}
$$

$$
\text{Probability constraints: } \sigma \ge 0, \quad \mathbf{1}^\top \sigma = 1.
\tag{9}
$$

## 3.2 EXISTENCE OF THE EQUILIBRIUM

Before a game begins, the selection of an equilibrium from a set of equilibria is essential to facilitate agreement among agents. Otherwise, a set of NE or CE solutions cannot be directly applied to the game because the agents lack the knowledge of how to coordinate in such a scenario. Therefore, ensuring the uniqueness of equilibrium through some rules is a common concern.

Maximum Welfare (Coarse) Correlation Equilibrium (MW(C)CE) is one of the well-known equilibrium concepts. Its optimization objective is typically the expectation of all players' payoffs, which often lacks a unique solution due to its linearity. Therefore MW(C)CE is often utilized as an evaluator of the quality of other equilibria. The concept of maximum entropy is extensively used in NE and CE, and they provide a criterion for selecting a unique solution within NEs and CEs (Ortiz et al., 2007; Balduzzi et al., 2018b). However, calculating entropy can be challenging, particularly in games with a large number of players and policies (actions). As discussed in Section 2.2, (C)CEs exist in the polytope, allowing us to employ any convex function to select a unique solution. MG(C)CE (Marris et al., 2021) uses a computationally straightforward quadratic function as the equilibrium selector.

The premise for the optimization problem to possess a unique solution is the convexity of the objective function. It appears that $-\frac{1}{2}\sigma^\top (I - \kappa \mathbf{f}\mathbf{f}^\top)\sigma$ is a convex function, yet the precise proposition hinges on an analysis of the matrix $\mathbf{f}\mathbf{f}^\top$. It is noteworthy that $\mathbf{f}\mathbf{f}^\top$ is a positive semi-definite matrix. Before discussing the existence of the equilibrium, we first present a proposition of $I - \kappa \mathbf{f}\mathbf{f}^\top$:

**Proposition 3.1.** $I - \kappa \mathbf{f}\mathbf{f}^\top$ *is an elementary matrix, and its eigen values are* $\{1, 1, 1, ..., 1 - \kappa \mathbf{f}^\top \mathbf{f}\}$.

Then we have proved that D(C)CE will have a unique solution when all eigen values of $I - \kappa \mathbf{f}\mathbf{f}^\top$ are greater than zero. A detailed description is shown in Theorem 3.2.

**Theorem 3.2** (Existence and Uniqueness). *D(C)CE provides a unique solution to the equilibrium selection problem and the solution exists when* $\kappa < \frac{1}{\mathbf{f}^\top \mathbf{f}}$.

## 3.3 CLOSED FORM SOLUTION OF D(C)CE

It is unrealistic to directly compute and optimize the Formula 7, so such problems are usually reformulated in the form of a Lagrangian function to obtain a closed form solution. We denote $\lambda_1$, $\lambda_2$, $\upsilon$ as the coefficients of (C)CE constraints, probability non-negativity constraints, and probability normalization constraints, respectively. The original objective can then be expressed as:

$$L = -\frac{1}{2}\sigma^\top(I - \kappa \mathbf{f}\mathbf{f}^\top)\sigma + \lambda_1^\top(A\sigma - \epsilon) - \lambda_2^\top \sigma + \upsilon(\mathbf{1}^\top \sigma - 1), \tag{10}$$

where $A = [A_1, A_2, ..., A_n]$ is the concatenation of all $A_i$ and $\mathbf{1}$ is a vector of ones. Based on the original objective, the simplified dual version can be formulated as:

$$\begin{aligned} L =& \frac{1}{2}\lambda_1^\top A F^{-1} C A^\top \lambda_1 + D^\top A^\top \lambda_1 + \frac{1}{2}\lambda_2^\top F^{-1} C \lambda_2 - D^\top \lambda_2 \\ & - \lambda_1^\top A F^{-1} C \lambda_2 - \epsilon^\top \lambda_1 + \frac{1}{2\mathbf{1}^\top F^{-1}\mathbf{1}}, \end{aligned} \tag{11}$$

where $F = I - \kappa \mathbf{f}\mathbf{f}^\top$, $F^{-1} = I - \frac{\kappa}{\kappa \mathbf{f}^\top \mathbf{f} - 1}\mathbf{f}\mathbf{f}^\top$, $D^\top = \frac{\mathbf{1}^\top F^{-1}}{\mathbf{1}^\top F^{-1}\mathbf{1}}$, $C = \mathbf{1}D^\top - I$. Then the optimal solution $\sigma^*$ can be expressed using the coefficient $\lambda$ of the two inequality constraints:

$$\sigma^* = F^{-1} C A^\top \lambda_1^* - F^{-1} C \lambda_2^* - D. \tag{12}$$

In the case of full-support, every value of the probability distribution is positive, that is, $\sigma > 0$. This implies that the dual variable $\lambda_2$ associated with the probability non-negativity constraint should be set to 0. Hence, the corresponding closed form solution is

$$\sigma^* = F^{-1} C A^\top \lambda_1^* - D. \tag{13}$$

In this manner, we can express the solution to Problem 7 analytically. Theorem 3.3 summarizes the description of solution. The number of $\lambda_1$ is no more than $\sum_i |\Pi_i|(\Pi_i - 1)$ in CEs and $\sum_i |\Pi_i|$ in CCEs, which restricts the number of full-support solutions and ensures that the solution is scalable to large-scale games.

**Theorem 3.3** (Closed Form Solution). *The closed form solution* $\sigma^*$ *to D(C)CE is formulated as:*

$$\text{General support: } \sigma^* = F^{-1} C A^\top \lambda_1^* - F^{-1} C \lambda_2^* - D, \tag{14}$$

$$\text{Full support: } \sigma^* = F^{-1} C A^\top \lambda_1^* - D. \tag{15}$$

There exists an $\epsilon$ that enables a fully-support D(C)CE solution. In principle, the magnitude of $\epsilon$ that allow full-support solutions is related to the type of full-support solutions, and Theorem 3.4 (Marris et al., 2021) describes this situation.

**Theorem 3.4** (Type of Full-support Solutions). *There is an* $\epsilon$ *such that a full-support D(C)CE solution exists. Specifically, the relationship between* $\epsilon$ *and the type of solutions is:*

1. *A uniform solution* $b$ *always exists when* $\max(Ab) \leq \epsilon$.

2. *The existing solution is non-uniform when* $\epsilon < \max(Ab)$.

---

**Algorithm 1** Diverse Correlated Oracle

---

**Input:** Initial policy populations: $\Pi_1^0, ..., \Pi_n^0 \leftarrow \{\pi_1^0\}, ..., \{\pi_n^0\}$
Compute payoff tensor: $G^0 \leftarrow \textbf{ER}(\Pi^0)$
Initialize meta strategies: $\sigma^0 \leftarrow \textbf{MS}(G^0)$
**for** $t \leftarrow \{1, ...\}$ **do**
    **for** $i \leftarrow \{1, ..., n\}$ **do**
        Training oracle $\pi_i^t$ and compute gap $\delta_i^t$ over $\sigma^{t-1}$: $\pi_i^t, \delta_i^t \leftarrow \textbf{BR}_i(\Pi^{0:t-1}, \sigma^{t-1})$
        Extend policy population $\Pi_i$: $\Pi_i^{0:t} \leftarrow \Pi_i^{0:t-1} \cup \{\pi_i^t\}$
    **end for**
    Compute payoff tensor at iteration $t$: $G^{0:t} \leftarrow \textbf{ER}(\Pi^{0:t})$
    Compute diversity over payoff and population at iteration $t$: $\text{Div}^t \leftarrow \textbf{DM}(G^{0:t}, \Pi^{0:t})$
    Compute meta strategies at iteration $t$: $\sigma^t \leftarrow \textbf{MS}(G^{0:t}, \text{Div}^t)$
    **if** $\sum_i \delta_i^t = 0$ **then**
        **break**
    **else**
        **continue**
    **end if**
**end for**
**Output:** Current meta-strategy for all players.

---

## 4 DIVERSE CORRELATED ORACLES

We present a new policy training approach, Diverse Correlated Oracles (DCO) in Algorithm 1, which integrates the concepts of Diverse (C)CE as outlined in Section 3. Although DCO simply solves the new equilibrium we introduced, it differs from other algorithms by assigning greater sampling probabilities to more diverse policies. This approach aids in promoting exploration within the policy space and narrowing the gap between the joint policy and the target equilibrium during the equilibrium-solving process.

Considering that Diverse Correlated Oracle is a iterative algorithm, the new policy solved by algorithm 1 at iteration $t$ is represented by $\pi_i^t$. The set of all policies obtained so far for players $i$ is expressed as $\Pi_i^{0:t}$, and the corresponding expected return (**ER**) is $G_i^{0:t}$. The diversity measure $\text{Div}^t$ of policy populations is solved by function $\textbf{DM}(G, \Pi)$. The Meta-Solver (**MS**), a function that takes in $G^{0:t}$ and $\text{Div}^t$, computes a joint policy $\sigma^t$ over the population $\Pi^{0:t}$. $\delta$ represents a gap between the joint policy $\sigma$ and the target equilibrium and plays an important role in the evaluation of the algorithm.

## 5 EXPERIMENTS

In order to verify that Diverse Correlated Oracle proposed in Section 4 can better approximate equilibrium through Diverse-CCE while solving $n$-player, general-sum games, we compare our method with the baseline algorithm JPSRO, which utilizes different Meta-Solvers, including Random Dirichlet, Uniform, MGCCE (Marris et al., 2021) and $\alpha$-Rank (Omidshafiei et al., 2019). The performance of various algorithms can be demonstrated by assessing the gap between the joint policy and the target equilibrium. The concept of gap is generalized from the exploitability (Lanctot et al., 2017) used in two-player zero-sum games. It measures the distance between a joint policy and (C)CE instead of NE in multi-player general-sum games. CCE Gap (Marris et al., 2021), which is evaluated under Maximum Welfare CCE, is formulated as:

$$
\begin{aligned}
\text{CCE Gap}(\sigma) &= \sum_i \left\lfloor \sum_{\pi \in \Pi} \sigma(\pi) \big[ u_i(\text{BR}_i, \pi_{-i}) - u_i(\pi) \big] \right\rfloor_+ \\
&= \sum_i \left\lfloor \mathbb{E}_{\pi \sim \sigma} \big[ u_i(\text{BR}_i, \pi_{-i}) - u_i(\pi) \big] \right\rfloor_+ .
\end{aligned}
\tag{16}
$$

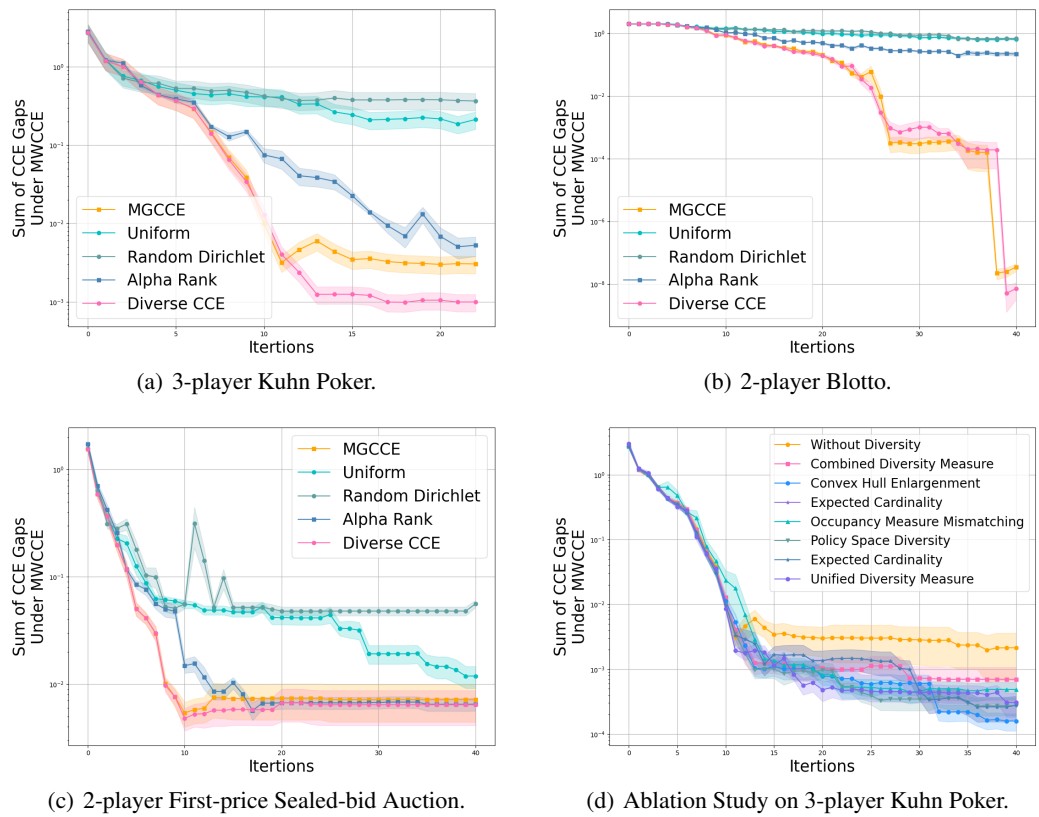

Figure 1: Gaps of the Meta CCE. Figure 1(a) to 1(c) demonstrates the advantages of DCO over MGCCE-based JPSRO, $\alpha$-Rank and some naive baselines. Figure 1(d) investigates the impact of different kinds of diversity measures on performance. **Sum of CCE Gaps Under MWCCE** is the sum of gaps between each player's joint policy and equilibrium. The smaller it is, the closer the joint policy found by the algorithm is to an exact equilibrium.

Our experimental results, conducted on 3-player Kuhn Poker, 2-player Blotto, and 2-player First-price Sealed-bid Auction games, provide empirical evidence supporting the advantages of DCO over alternative approaches. All games are included in OpenSpiel (Lanctot et al., 2019). As discussed in Theorem 3.2, the weight $\kappa$ of the diversity term should be smaller than $\frac{1}{7}\mathbf{f}^\top\mathbf{f}$. In addition, considering that the size of the diversity vector $\mathbf{f}$ is equal to $\prod_{i=0}^{n}|\Pi_i|$, in order to normalize the matrix $\mathbf{ff}^\top$, we set $\kappa = \frac{\beta}{\prod_{i=0}^{n}|\Pi_i|} < \frac{1}{\mathbf{f}^\top\mathbf{f}}$, where $\beta$ is selected appropriate through extensive hyper-parameter tuning. We conduct multiple different random experiments for each game. We also selected different diversity measures for ablation research on 3-player Kuhn Poker.

**Kuhn Poker**. It is a simple 2-player zero-sum imperfect-information game developed by Harold W. Kuhn (Kuhn, 1950) and has been extended to the **3-player** situation (Szafron et al., 2013). In this game, the total number of cards is 10, of which each player holds one card. The player chooses to bet or check when there is no outstanding bet; if there is an outstanding bet, the player folds or fold. When at least one player calls or everyone checks, showdown of all players begins; otherwise the bettor wins. The results in Figure 1(a) show that the CCE Gap under DCO is 67.31% lower than that under MGCCE and 81.12% lower than that under $\alpha$-Rank. Our method exhibits convergence after approximately 20 iterations, leading to an early termination of the experiment. It is worth noting that the convergence speed of $\alpha$-Rank is slower in comparison to DCO. Furthermore, $\alpha$-Rank incurs a significantly higher computational complexity. When iterating for the same number of steps, $\alpha$-Rank requires approximately 5.5 times more time compared to DCO.

**Blotto**. Blotto (Borel, 1953; Roberson, 2006) is a classic 2-player zero-sum imperfect-information game in which two players allocate resources in multiple areas. In this game, both players have a budget of coins, which are allocated to a fixed number of areas simultaneously. When a player places the most coins in an area, the player wins that area. The final winner is the player who wins the most areas (Ferdowsi et al., 2018). The performance comparison between different algorithms is shown in Figure 1(b). The results demonstrate that our method achieves the smallest gap, accounting for only 20.67% of the CCE Gap under MGCCE and being approximately 8 orders of magnitude smaller than the CCE Gap under $\alpha$-Rank.

**First-price Sealed-bid Auction (FPSBA)**. It is an multi-player general-sum imperfect-information game, also known as a blind auction (Bhattacharya et al., 2018). All bidders submit sealed bids simultaneously and are unaware of the bids of other participants (Krishna, 2009; McAfee & McMillan, 1987). Ultimately the highest bidder pays. This is a classic Bayesian game (Harsanyi, 1967; Zamir, 2020) in which one player does not know the payoffs of the other players. Solving the equilibrium is challenging because bidders' valuations of items for sale are difficult to predict. As depicted in Figure 1(c), DCO in 2-player FPSBA attains a performance level nearly equivalent to that of $\alpha$-Rank, while exhibiting approximately a 10% lower gap compared to MGCCE.

**Ablation Study.** We studied the impact of different diversity measures on convergence at 3-player Kuhn Poker. By changing the diversity measure in Figure 1(d), we report the gap evaluated by MWCCE. The overall performance of RD surpasses that of BD. Specifically, comparing the performance of the two Behavior Diversities, Policy Space Diversity outperforms Occupancy Measure Mismatching; while in RD, Convex Hull Enlargement yields the smallest CCE Gap. The second-best performing measure is Expected Cardinality, which demonstrates performance very similar to that of Policy Space Diversity. The Unified Diversity Measure falls between the two Behavior Diversities in terms of performance. RD offers a more direct reflection of payoff dynamics, and our evaluation metric, CCE Gap, is constructed primarily based on payoffs rather than state-action distribution. Consequently, RD outperforms BD. In RD, Convex Hull Enlargement performs best, probably due to its ability to provide a more comprehensive description of policy dynamics by measuring the differences of each policy from others, instead of computing diversity over the entire population, as done by Expected Cardinality and Unified Diversity Measure.

## 6 CONCLUSION

This paper explores a novel open-ended learning framework for solving multi-player general-sum games, which incorporates diversity measures into solving equilibria for meta-games. To this end, we extend the concept of policy diversity measure in 2-player zero-sum games to $n$-player general-sum games, and define a new correlated equilibrium, Diverse (C)CE, with the objective of maximizing the sum of each player's policy diversity expectations. Additionally, we introduce a new policy training algorithm called Diverse Correlated Oracle to approximate Diverse (C)CE. Empirical results demonstrate the advantages of our approach when applied to a variety of multi-player general-sum games.

Furthermore, we discuss the limitations of our method. Firstly, one limitation is that the weight $\kappa$ of the diversity expectation term is manually adjusted. Although we associate $\kappa$ with the overall size of the policy population, manual adjustments remain necessary. We believe that this weight should be associated to the specific structure of the game and the upper bound of the diversity measure. In future work, we will conduct in-depth analyses of the structures of typical games, such as characteristics akin to the non-transitiveness observed in two-player games. These analyses will inform the determination of a reasonable weight. Similar ideas have been proposed and implemented in single-agent reinforcement learning, where they learned discount rates in this way (Xu et al., 2018). Additionally, the performance of our method in $n$-player general-sum games has not reached a satisfactory level. To be more specific, while our approach does outperform existing methods in terms of gap, it does not significantly contribute to the enhancement of social welfare. In general, real-world scenarios are better represented by multi-player general-sum games. In these games, players aim not only to coordinate their policies or actions within a unified joint policy but also seek to maximize overall social welfare. Hence, our future research should prioritize enhancing the performance of DCO with regard to social welfare.

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

APPENDIX

# A   FULL PROOF OF PROPOSITIONS AND THEOREMS

## A.1   PROOF OF PROPOSITION 3.1

**Proposition 3.1.** $I - \kappa \mathbf{f}\mathbf{f}^\top$ *is an elementary matrix, and its eigen values are* $\{1, 1, 1, ..., 1 - \kappa \mathbf{f}^\top \mathbf{f}\}$.

*Proof.* First, the definition of elementary matrix (Horn & Johnson, 2012) is given: Assume $u, v \in C^n, \kappa \in C$, then $E(u, v, \kappa) = I - \kappa u v^\top$ is an elementary matrix, where $C^n$ is the vector space and $C$ is the set of real numbers.

Then we let $u = \mathbf{f}, v = \mathbf{f}$, obviously $E(\mathbf{f}, \mathbf{f}, \kappa) = I - \kappa \mathbf{f}\mathbf{f}^\top$ is an elementary matrix.

Regarding eigenvalues, an elementary matrix $E(u, v, \kappa)$ has the following properties (Horn & Johnson, 2012):

**Proposition A.1.** *The eigenvalues of* $E(u, v, \kappa)$ *are* $\{1, 1, 1, ..., 1 - \kappa v^\top u\}$.

Based on A.1, the eigenvalues of $I - \kappa \mathbf{f}\mathbf{f}^\top$ can be formulated as $\{1, 1, 1, ..., 1 - \kappa \mathbf{f}^\top \mathbf{f}\}$.

## A.2   PROOF OF THEOREM 3.2

**Theorem 3.2** (Existence and Uniqueness). *D(C)CE provides a unique solution to the equilibrium solution problem and the solution exists when* $\kappa < \frac{1}{\mathbf{f}^\top \mathbf{f}}$.

*Proof.* When the objective function of the optimization problem is convex quadratic and the constraints are affine, the problem is called quadratic programming, and there is a unique solution to the quadratic programming problem (Boyd & Vandenberghe, 2004).

The objective function of D(C)CE, $-\frac{1}{2}\sigma^\top (I - \kappa \mathbf{f}\mathbf{f}^\top)\sigma$, is a quadratic. To make it a convex quadratic, the matrix $I - \kappa \mathbf{f}\mathbf{f}^\top$ must be positive definite. A square matrix is a positive definite matrix if all its eigenvalues are positive (Strang, 2022). Based on Proposition 3.1, if $1 - \kappa \mathbf{f}^\top \mathbf{f} > 0$ or $\kappa < \frac{1}{\mathbf{f}^\top \mathbf{f}}$, then all eigenvalues of $I - \kappa \mathbf{f}\mathbf{f}^\top$ are positive. In addition, the constraints of Diverse (C)CE are all linear affine. To sum up, when $\kappa < \frac{1}{\mathbf{f}^\top \mathbf{f}}$, there is a unique solution to Diverse CCE.

## A.3   PROOF OF THEOREM 3.3

**Theorem 3.3** (Closed Form Solution). *The closed form solution* $\sigma^*$ *to D(C)CE is formulated as:*

$$\text{General support: } \sigma^* = F^{-1}CA^\top \lambda_1^* + F^{-1}C\lambda_2^* - D,$$

$$\text{Full support: } \sigma^* = F^{-1}CA^\top \lambda_1^* - D,$$

where $F = I - \kappa \mathbf{f}\mathbf{f}^\top$, $F^{-1} = I - \frac{\kappa}{\kappa \mathbf{f}^\top \mathbf{f} - 1}\mathbf{f}\mathbf{f}^\top$, $D^\top = \frac{\mathbf{1}^\top F^{-1}}{\mathbf{1}^\top F^{-1}\mathbf{1}}$, $C = \mathbf{1}D^\top - I$ and $\mathbf{1}$ is a vector of ones.

*Proof.* We start with the primal Lagrangian form:

$$L = -\frac{1}{2}\sigma^\top (I - \kappa \mathbf{f}\mathbf{f}^\top)\sigma + \lambda_1^\top (A\sigma - \epsilon) - \lambda_2^\top \sigma + \upsilon(\mathbf{1}^\top \sigma - 1),$$

Then we take derivatives with respect to the primal variables $\sigma$, and make them equal to zero.

$$\frac{\partial L}{\partial \sigma} = F\sigma + A^\top \lambda_1 - \lambda_2 + \upsilon \mathbf{1} = 0 \Rightarrow$$

$$\sigma^* = -F^{-1}(A^\top \lambda_1 - \lambda_2 + v\mathbf{1})$$

Then $\sigma^*$ can be substituted into the Lagrangian function.

$$
\begin{aligned}
L = & -\frac{1}{2}(\lambda_1^\top A F^{-1} A^\top \lambda_1 - \lambda_1^\top A F^{-1}\lambda_2 + \lambda_1^\top A F^{-1}v\mathbf{1} - \lambda_2^\top F^{-1}A^\top \lambda_1 \\
& + \lambda_2^\top F^{-1}\lambda_2 - \lambda_2^\top F^{-1}v\mathbf{1} + \mathbf{1}^\top v F^{-1}v\mathbf{1} + \mathbf{1}^\top v F^{-1}A^\top \lambda_1 - \mathbf{1}^\top v F^{-1}\lambda_2) \\
& - \lambda_1^\top \epsilon - v
\end{aligned}
$$

We take derivatives with respect to $v$ and then make the derivatives equal to zero.

$$\frac{\partial L}{\partial v} = -\frac{1}{2}(\lambda_1^\top A F^{-1}\mathbf{1} - \lambda_2^\top F^{-1}\mathbf{1} + \mathbf{1}^\top F^{-1}A^\top \lambda_1 - \mathbf{1}^\top F^{-1}\lambda_2 + 2\mathbf{1}^\top F^{-1}\mathbf{1}v) - 1 = 0 \Rightarrow$$

$$v^* = -\frac{\mathbf{1}^\top F^{-1}A^\top \lambda_1 - \mathbf{1}^\top F^{-1}\lambda_2 + 1}{\mathbf{1}^\top F^{-1}\mathbf{1}}$$

We then substitute $v^*$ into the Lagrangian function $L$.

$$
\begin{aligned}
L = & \frac{1}{2}\lambda_1^\top A F^{-1}(\frac{\mathbf{1}\mathbf{1}^\top F^{-1}}{\mathbf{1}^\top F^{-1}\mathbf{1}} - I)A^\top \lambda_1 + \frac{\mathbf{1}^\top F^{-1}A^\top \lambda_1}{\mathbf{1}^\top F^{-1}\mathbf{1}} \\
& + \frac{1}{2}\lambda_2^\top F^{-1}(\frac{\mathbf{1}\mathbf{1}^\top F^{-1}}{\mathbf{1}^\top F^{-1}\mathbf{1}} - I)A^\top \lambda_2 - \frac{\mathbf{1}^\top F^{-1}\lambda_2}{\mathbf{1}^\top F^{-1}\mathbf{1}} \\
& - \lambda_1^\top A F^{-1}(\frac{\mathbf{1}\mathbf{1}^\top F^{-1}}{\mathbf{1}^\top F^{-1}\mathbf{1}} - I)\lambda_2 - \lambda_1^\top \epsilon + \frac{1}{2 \times \mathbf{1}^\top F^{-1}\mathbf{1}}
\end{aligned}
$$

We let $F = I - \kappa\mathbf{f}\mathbf{f}^\top$, $F^{-1} = I - \frac{\kappa}{\kappa\mathbf{f}^\top\mathbf{f}-1}\mathbf{f}\mathbf{f}^\top$, $D^\top = \frac{\mathbf{1}^\top F^{-1}}{\mathbf{1}^\top F^{-1}\mathbf{1}}$ and $C = \mathbf{1}D^\top - I$. The Lagrangian function can be simply formulated.

$$
\begin{aligned}
L = & \frac{1}{2}\lambda_1^\top A F^{-1}C A^\top \lambda_1 + D^\top A^\top \lambda_1 \\
& + \frac{1}{2}\lambda_2^\top F^{-1}C A^\top \lambda_2 - D^\top \lambda_2 \\
& - \lambda_1^\top A F^{-1}C\lambda_2 - \epsilon^\top \lambda_1 + \frac{1}{2 \times \mathbf{1}^\top F^{-1}\mathbf{1}}
\end{aligned}
$$

Next we substitute $v^*$ into $\sigma^*$ to get the closed form solution:

$$
\begin{aligned}
\sigma^* &= -F^{-1}(A^\top \lambda_1 - \lambda_2 + v^*\mathbf{1}) \\
&= F^{-1}C A^\top \lambda_1^* - F^{-1}C\lambda_2^* - D.
\end{aligned}
$$

For the case where $\sigma$ has full support, $\lambda_2 = 0$ holds, because any $\sigma \geq 0$ constraint has no effect. Then the closed form solution in full-support cases is

$$\sigma^* = F^{-1}C A^\top \lambda_1^* - D.$$

## A.4 PROOF OF THEOREM 3.4

**Theorem 3.4** (Type of Full-support Solutions). *There is an $\epsilon$ such that a full-support D(C)CE solution exists. Specifically, the relationship between $\epsilon$ and the type of solutions is:*

1. *A uniform solution $b$ always exists when $\max(Ab) \leq \epsilon$.*

2. *The existing solution is non-uniform when $\epsilon < \max(Ab)$.*

*Proof.* Recalling the optimization objective $-\frac{1}{2}\sigma^\top(I - \kappa\mathbf{ff}^\top)\sigma$, it is obvious that when $\sigma = b$, the optimization objective reaches its maximum value. Additionally, $\sigma$ must satisfy the linear constraint $A\sigma \le \epsilon$. Therefore, if $\max(Ab) > \epsilon$, then $\sigma$ cannot be uniform. On the contrary, when $\max(Ab) \le \epsilon$, a uniform solution $b$ must exist.

