# OpenReview forum: "Open-Ended Learning in General-Sum Games: The Role of Diversity in Correlated Equilibrium"
_ICLR.cc/2024/Conference — Submitted to ICLR 2024_

### Official Review · Reviewer_4LQy · 2023-10-12

**Soundness:** 2 fair
**Presentation:** 1 poor
**Contribution:** 2 fair
**Rating:** 3
**Confidence:** 3

**Summary:**

The paper introduces a new meta-solver for Joint PSRO (JPSRO) for use in computing CCE and CE. Their meta-solver includes a term for the "diversity" of the resulting strategy. Empirical experiments seem to suggest that their meta-solver yields faster convergence at least in some games.

**Strengths:**

The concept of using policy diversity as a metric in selecting a meta-strategy for PSRO is interesting. It seems to have a positive effect on the convergence rate in some games.

**Weaknesses:**

My main concern is the quality of the writing. I had trouble understanding the main ideas of the paper, mainly because there were many confusing definitions and notational jumps. Here are some concerns, in no particular order.

1. In 2.1, the definition of a "game" seems to be a complete-information, alternating-move sequential game. But the paper is general to incomplete-information or simultaneous-move settings as well, so I think that the definition of game should be changed to accomodate that. In fact, since the paper doesn't care about the sequentiality at all, why not just use normal-form games throughout?

1. The citation of [Farina et al 2020, Marris et al 2022] for correlated equilibrium is strange---especially the former, as the concept it discusses isn't CE. For CE I believe the correct citation is [Aumann 1974].

1. The definition of $\epsilon$-CE used here is a bit strange. Indeed, it seems to imply that the uniform-random policy is a $1/N$-CE, where $N = \min_i |\Pi_i|$ (which could be very large), and that not every $\epsilon$-CE is an $\epsilon$-CCE. Perhaps it would be better to define $\epsilon$-CE with the set of swap deviations instead of the set of internal deviations (see e.g. [Blum & Mansour 2007]). But also this doesn't seem to matter much to the rest of the paper, so I mostly disregard it.

1. In the diversity metric definitions (3) and (4):

    a. the metric seems to depend on the ordering in which the profiles are added (i.e., on which strategy is the new one). Perhaps then the diversity metric should be expressed as a two-argument function, $\text{Div}(\Pi_i, \pi_i')$, so that this dependence is clear.

    b. The definition of occupancy measure should be stated in the text.

    c. Since Nash equilibria are not generally unique, (4) is not well-defined until a specific NE is selected. This should be explicitly stated and discussed.

    d. It is unclear how the definition $\text{Div}(\cdot)$ relates to the function $f(\cdot)$ used later in the paper. In particular, it appears as if the diversity metric $\text{Div}(\cdot)$ is built to affect how *new policies* are introduced (i.e. to affect the best-response oracle, which is the thing picking the $pi_i'$), whereas the rest of the paper concerns the meta-solver. How are these related? This to me is a crucial point, because $f$ is used liberally in the remainder of the paper.

The experimental evaluation is also not very convincing. On the single example in which the authors do a comprehensive test against other metrics, the proposed method, which is a combination of two prior methods, performs second-worst, being outperformed by both of the prior methods individually.

A Blum & Y Mansour. "From External to Internal Regret", JMLR 2007

**Questions:**

1. (from 4(d) above) How does the definition $\text{Div}(\cdot)$ relate to the function $f(\cdot)$ used later in the paper?

1. For Prop 3.1, what's an "elementary matrix"? $I - \kappa \mathbf{ff}^\top$ does not seem to satisfy the [definition I am familiar with](https://en.wikipedia.org/wiki/Elementary_matrix)

1. Why is it "unrealistic" to solve (7-9)? If the game has a manageable size, it's is just a convex program.

1. Doesn't Theorem 3.4 actually hold for all $\epsilon > 0$?

---

> ### Author Response · Authors · 2023-11-17
> **Response to Reviewer 4LQy**
>
> Thank you for the reviewer's suggestions. We appreciate the opportunity to discuss the theoretical details with the reviewer.
>
> >(from 4(d) above) How does the definition Div(⋅) relate to the function f(⋅) used later in the paper?
>
> I apologize for any confusion in our explanation at this point. The meanings of $\text{Div}(⋅)$ and $\textbf{f}(⋅)$ are indeed consistent. The difference lies in the usage: $\text{Div}(⋅)$ is employed in the algorithm's pseudocode to represent the matrix formed by the diversities of all policies for all players, while $\textbf{f}(⋅)$ is used in the equilibrium definition, signifying that diversity is a function with policies as variables.
>
> >For Prop 3.1, what's an "elementary matrix"? I−\kappa ff^\top does not seem to satisfy the definition I am familiar with.
>
> In some books about matrix theory, elementary matrices are defined as follows: The general form of an elementary matrix is given by $E(u, v, \kappa)=I-\kappa uv^\top$, where $u,v\in C^n$ and $\kappa\in C$. All elementary transformations can be represented in the form of $E(u, v, \kappa)$. Upon further investigation, we found that a more widely accepted definition for such matrices is "rank-one update," i.e., $E(u, v, \kappa)$ is a rank-one update of the identity matrix $I$. We will correct this definition in the paper.
>
> >Why is it "unrealistic" to solve (7-9)? If the game has a manageable size, it's is just a convex program.
>
> Thank you for the clarification. Our expression here was a bit imprecise. What we intended to convey is that in most scenarios, directly solving equations (7-9) is not practical. Therefore, in convex optimization literature, it is common to transform the problem into solving the Lagrangian function. Typically, the size of the game can be manageable, but the size of the population may not be. Of course, in the case of toy games, the problem can be solved directly.
>
> >Doesn't Theorem 3.4 actually hold for all $\epsilon>0$?
>
> Yes, Theorem 3.4 actually holds for all $\epsilon>0$. We provide an explanation for this in the appendix. Recalling the optimization objective $−\frac{1}{2}\sigma^\top(I − \kappa \textbf{f}\textbf{f}^\top)\sigma$, it is obvious that when $\sigma=b$, the optimization objective reaches its maximum value. Additionally, $\sigma$ must satisfy the linear constraint $A\sigma\leq\epsilon$. Therefore, if $\max(Ab) > \epsilon$, then $\sigma$ cannot be uniform. On the contrary, when $\max(Ab) \leq\epsilon$, a uniform solution $b$ must exist.

---

> > ### Comment · Reviewer_4LQy · 2023-11-18
> >
> > As I mentioned in my review, my core problem with the paper was a concern about clarity, and the paper is still generally too unclear for me to understand the main ideas. I am clearly also not the only reviewer with these concerns. I would advise a major revision in which the authors take into account the clarity concerns from myself and the other reviewers, in order for the main ideas of the paper to be more understandable. I will maintain my score.
> >
> > If Div and f are the same function, the same notation should be used for both. That would be far less confusing.
> >
> > The definition that I know for "elementary matrix" is that an elementary matrix is one that is created by performing a single elementary row operation (switching, multiplication, or addition of rows) on the identity matrix. The difference in definition should be clarified.
> >
> > On (7-9), if I understand correctly, that equation is only ever being solved over the meta-game. The meta-game will be a very small normal-form game, in which case (7-9) is a convex program of reasonably small size, which is probably realistically solvable. I would advise that the authors try a generic off-the-shelf convex optimizer (such as Gurobi) to solve (7-9).

---

### Official Review · Reviewer_SpBs · 2023-10-27

**Soundness:** 2 fair
**Presentation:** 1 poor
**Contribution:** 1 poor
**Rating:** 3
**Confidence:** 3

**Summary:**

This paper addresses the problem of solving for correlated equilibrium in n-player general-sum games. They accomplish this through the Policy-Space Response Oracles (PSRO) framework that solves a game by proxy through an empirical game. From this, they study the role of different diversity-based exploration methods in improving the quality of the solution computed. To do this they propose increasing the sampling probability assigned to "diverse" policies in the solution. They evaluate their method on a few OpenSpiel 2/3-player games.

**Strengths:**

- Studies an important problem which is the role of diversity in strategy exploration methods (a subroutine of empirical game solving).
- Includes an ablation study that covers the space of both reward and behavioral diversity measures.
- Theoretically, and empirically, show that their solution can be computed analytically.

**Weaknesses:**

- It's not clear to me why we should want to "increase the sampling probability assigned to diverse policies" nor that this is well defined.
  - The diverse policies are not necessarily profitable in equilibrium.
  - If our population is diverse then this suggests we want a uniform distribution. Otherwise, the population isn't diverse, and then increasing the sampling probabilities of diverse policies would reduce return.
  - How would this claim apply to deterministic versus stochastic policies? Including purification of policy subsets?
- The contributions of this paper appear marginal, as it suggests using existing diversity measures to solve for maximum welfare coarse correlated equilibrium (MWCCE).
  - The paper frames itself as solving for CE and CCE generally but only has an evaluation for MWCCE.
  - The interesting ablation study of different diversity metrics is shown only on a single game (3-player Kuhn poker).
  - The empirical performance of "Diverse CCE" only shows a benefit over a baseline in one of the three games studied.

**Questions:**

- "the isolation diversity term is meaningless" What do the authors mean by this? Could they define this precisely?
- Why is NE not included as a baseline meta-strategy solver?
- How does the ablation study perform on the other games?
- "Given that diversity has the ... and being computationally tractable." Diversity is not generally guaranteed to be tractable.
- How do the methods compare if you're interested in finding an equilibrium as opposed to the maximum welfare equilibrium?
- Some of the citations are confusing, for example Correlated Equilibrium is credited by two papers that came out after 2020.

Nits:
- frameowrk --> framework
- Issues with spacing before citations and after parenthesis throughout.

---

> ### Author Response · Authors · 2023-11-17
> **Response to Reviewer SpBs**
>
> We appreciate the reviewer's comments and suggestions.
>
> >"the isolation diversity term is meaningless" What do the authors mean by this? Could they define this precisely?
>
> As mentioned in the common response, some diversity definitions focus on the difference between an individual policy and all other policies within the population. Therefore, "the isolation diversity" refers to the scenario where we adhere to this definition and calculate diversity only for a specific policy within the population, without computing the diversity for all policies. This isolated diversity can only reflect the differences between the corresponding policy and all other policies in the population, failing to capture the overall diversity of the entire population. Hence, it is meaningless. We appreciate your suggestion, and we do acknowledge the need to provide an explanation for these points in the paper.
>
> >Why is NE not included as a baseline meta-strategy solver?
>
> For multi-player games, Nash Equilibrium (NE) is not unique, and its computational cost is substantial. Therefore, it is not suitable as a baseline meta-strategy solver, which is why JPSRO utilizes (C)CE instead of NE. However, in the case of 2-player games, the reviewer's suggestion is reasonable, and indeed, we should consider NE as a baseline meta-strategy solver for 2-player games.
>
> >How does the ablation study perform on the other games?
>
> We will supplement the ablation study performed on the other games in our paper.
>
> >"Given that diversity has the ... and being computationally tractable." Diversity is not generally guaranteed to be tractable.
>
> To our knowledge, diversity is commonly defined in computationally tractable forms, such as vector norms, matrix norms, and distribution divergences.
>
> >How do the methods compare if you're interested in finding an equilibrium as opposed to the maximum welfare equilibrium?
>
> In game theory, there is a widespread interest in equilibria that maximize the total payoff for all players, known as the maximum welfare equilibrium. This represents one of the fundamental questions in game theory research. Therefore, in our work, we have also chosen the maximum welfare equilibrium as an evaluation metric, as it provides an objective measure for assessing outcomes.
>
> >Some of the citations are confusing, for example Correlated Equilibrium is credited by two papers that came out after 2020.
>
> Thank you for your suggestion. We will correct this section accordingly.

---

> > ### Comment · Reviewer_SpBs · 2023-11-20
> >
> > Thank you for taking the time to address some of my concerns. Unfortunately, my main concern is still unresolved for myself and my fellow reviewers: the motivation and justification of the primary hypothesis are insufficient. For this reason, my overall evaluation of the manuscript remains largely unchanged. Specific comments and questions follow.
> >
> >
> > >  Therefore, a reasonable approach is to provide greater sampling opportunities for opponent policies that are farther from the population (i.e., more diverse), which can avoid repeatedly facing similar opponent policies in different training rounds, thereby enhancing the adaptability of policies more quickly. However, currently, no work has considered this aspect.
> >
> > I don't think it's fair to claim that increasing the probability of sampling diverse coplayers is novel to this work. One such direct example is Rectified Nash, but I would make arguments that FictiousPlay, History-Aware Double Oracle, Mixed-Opponents, NeuPL, etc. all do this in essence.
> >
> >
> > > We will supplement the ablation study performed on the other games in our paper.
> >
> > I'm glad to hear that additional analysis will be conducted. These should greatly aide the quality of the manuscript.
> >
> >
> > > To our knowledge, diversity is commonly defined in computationally tractable forms, such as vector norms, matrix norms, and distribution divergences.
> >
> > I think this is only a fair claim to make for response-diversity terms. Computing state-action occupancy distributions for a policy can be computationally intractable, generally.
> >
> >
> > > Therefore, in our work, we have also chosen the maximum welfare equilibrium as an evaluation metric
> >
> > I agree that this is a fine definition of equilibrium and a reasonable one to choose (especially since it resolves the selection problem). However, the premise of the paper is more general than this claim. I would suggest refining the scope of the paper throughout if the goal is only MW(C)CE, or suggest studying (C)CE generally.

---

### Official Review · Reviewer_EpNp · 2023-10-31

**Soundness:** 3 good
**Presentation:** 2 fair
**Contribution:** 2 fair
**Rating:** 3
**Confidence:** 2

**Summary:**

This paper defines a new metric for policy diversity in multi-player games, then defines a new equilibrium concept (Diverse [Coarse] Correlated Equilibrium), then proposes an extension of population-based methods PSRO / JPSRO that presumably finds this equilibrium (Diverse Correlated Oracles).

Theory is given for the uniqueness of such an equilibrium, and a closed-form solution to find it.

Experiments are performed on a few general-sum or multi-player games, showing the CCE gap of the novel algorithm versus JPSRO with various diversity metrics.

**Strengths:**

The research direction is interesting: investigating definitions of diversity in multi-player games, and using them to define equilibrium concepts and using them with a PSRO-like algorithm. The theory given seems sound (although I did not check carefully).

**Weaknesses:**

In its current state, this paper is not ready for publication. Most of the issues are in the writing of the paper. In my reading, I was unable to find explanations for key parts of the algorithm. There are also some minor issues: it's not intuitive to me why the definition of diversity expectation would be good, which would be fine if the experiments convincingly showed that the algorithm is good, but they don't. (Maybe the definition of diversity expectation is a good one, but the paper didn't show me that it is.)

- The algorithm is not clearly explained: I feel like this paper is missing a paragraph or a section, because Section 4 and Algorithm 1 should explicate the algorithm. However, to my reading, they do not. The paper does not explain what MS is, which is the most important part of the paper. (Without stating the specific choice of MS, Algorithm 1 seems to be equivalent to PSRO.) I infer that MS does the computation described in Section 3.3, but this does not seem to be explicitly stated anywhere! And even if I make this inference, I don't know whether it is Equation 14 (General support) or Equation 15 (Full support) which is being calculated. The paper also does not explain what DM is, although I infer that it is the diversity measure from Equation 6.

- However, this also seems undefined! Section 3.1 explains how sum of diversity expectations is calculated based on each player's diversity measure $f_{i|l,m,...q}$, which is defiend in terms of $f$. However, as far as I can see, the choice of $f$ used in the experiments was never described! (I cannot even infer the choice of $f$, since Section 2.4 describes two different diversity measures, and states that this work uses a combination of them, with the method of combination left undefined.)

- Presumably the output of DCO is supposed to be a D(C)CE, but this is never stated. It should be stated. It should also be stated and proved that the termination condition given in Algorithm 1 leads to the output being a D(C)CE.

- I don't entirely understand the sentence in Section 3.1: "Inspired by JPSRO, we use a quadratic function as the optimization objective". If my interpretation is correct, JPSRO uses the Gini impurity as an objective, which is quadratic. Here, I don't see why you would be inspired to maximize $(f^\top \sigma)^2$ -- aren't both of those vectors non-negative, so maximizing their square is equivalent to maximizing the original thing?

- It's stated in Section 3.1 (and throughout the paper) that "our objective it to increase the sampling probability assigned to diverse policies. This is achieved by maximizing the sum of each player's policy diversity expectations, thereby ensuring that the more exploratory policies play a more significant role in approximating the full game (C)CE." I don't immediately see why this objective is achieved by maximizing the sum of each player's policy diversity expectations. Maybe I am misunderstanding, but as defined, it seems that each $f_{i|l,m,...q}$ is dependent on the set of player $i$ policies, but is not dependent on their distribution over those policies. It would be helpful for the paper to explain or give intuition for why the policy diversity expectations achieve this goal.

- On that note, it would be good for the paper to include qualitative examples of things like the policy diversity expectation value for different strategy profiles, and especially for the difference in distributions over a given set of strategy profiles for D(C)CE vs. MG(C)CE.

- Results are not convincingly strong. Only in 3-Player Kuhn Poker is the novel method clearly better. 2 of the 3 experiment games are 2-player, not multi-player.

Minor:
- the first paragraph of Section 3 mentions "MGCE" but this is not defined anywhere.
- Game notation (Section 2.1) could use work. Probably should clarify in the first sentence that this describes sequential games. Policies are not defined. State transitions are not defined. In the description of meta-game, should it say that "a meta-strategy is used to denote the probability distribution over policy *profiles*" instead of "over policies"? And perhaps a more suitable name for "meta-strategy" would be "joint meta-strategy"?
- Notation in Section 2.2 could also be more clear. It could be specified what the summation in Inequality (1) is over. Sigma is defined in 2.1 using brackets but is used in 2.2 with parentheses. Underneath it says it can be written as $A_i \sigma$ but neither of those symbols are defined (it is not explicated anywhere that they are vectors). I would appreciate a citation for the statement that the maximum sum of social welfare under CEs is greater than or equal to that under NEs. Should it say that "The NE *are* located on the boundary" instead of "The NE *is* located on the boundary"? In the definition for NE, the symbol $\sigma(\pi_i)$ is used, and although it's clear to someone familiar with the topic what this means, it's not defined anywhere and could be confusing for some readers. In the definition for CCE, should it say "only consider deviating *before*" instead of "only consider deviating *until*"? Also, this "recommended policy" is referenced to define CCEs, but is not previously mentioned in the definition of CEs, so the reader doesn't know what a "recommended policy" is. For the definition of CCEs (inequality 2), the foralls should be clearly restated instead of implied, otherwise the definition of $i$ is not clear.
- Section 2.3 uses the term "Meta-Solver (MS)" but doesn't describe what it is or give any examples (e.g. Nash Equilibrium solver). The symbol $u^*_i$ is used but not defined anywhere.
- Section 3.2 is titled "Existence of the Equilibrium" but it seems to me to concern the uniqueness of the equilibrium and not the existence of the equilibrium.
- Section 4: "it differs from other algorithms by assigning greater sampling probabilities to more diverse policies": could this be made more precise or expanded upon? What is a "more diverse policy"? When does it assign greater sampling probability? (in the meta-solver?)
- Section 6: The sentence "In [multi-player general-sum games], players aim not only to coordinate their policies or actions within a unified join policy but also seek to maximize overall social welfare" seems wrong to me. That may be the players' goal, or it may be the algorithm designer's goal, but generally players just care about their own utility, right? Unless the setting is intended to be fully cooperative.


Nitpicks / typos:

- Abstract: "The primary in this work"
- Introduction: "frameowrk"
- Quotation marks in Section 2.1 are both in the same direction for "game of games"
- Section 2.1: "used to denoted" should be "used to denote"
- Section 2.2: it's weird that the original Aumann paper is not cited here for the definition of CE (I know it was cited earlier, in the Introduction, but should it be cited here?)
- Section 2.4: Perhaps rearrange the first sentence so it says "... the diversity of player $i$'s population $\Pi_i ...$, given the opponent's policy $\pi^l_{-i}$  is denoted by ..." to emphasize that the diversity is of a population and conditional on an opponent's policy?
- Section 3: grammar - "The sampling probability of the policy is closer to the uniform distribution, the entropy or Gini impurity is larger." Also, the "sampling probability of a policy" is referred to a few times but as far as I can tell it's not precisely defined what it means.
- Section 3: should "regular term" be "regularization term"? This would also benefit from a quick explanation of "policy exploitability (PE)"
- should "the corresponding expected return (ER)" be "the corresponding expected returns (ER)"?
- Section 5 (page 8): "the weight K of the diversity term should be smaller than ..." the fraction has a typo?
- Figure 1 x-axis should be "Iterations" not "Itertions"

**Questions:**

- The explanation of the metrics and equilibrium concept and algorithm implies that they can be used for both correlated as well as coarse correlated equilibria. However, the experiments seem to all use CCE versions and measure the CCE gap. Can the experiments be performed for CE as well?

- Throughout the text, it's stated that we want to assign higher sampling probabilities to "more diverse policies". But what does it mean for a policy to be "more diverse" than another? Isn't diversity defined for a set of policies, not for an individual policy?

- I don't understand the (C)CE in Equation 8. When this is for CCE, what is $A_i$?

- Section 3.2 says that "calculating entropy can be challenging" -- why is that? Isn't it just $O(|N| \times |\Pi|)$ where $|\Pi|$ is the size of each player's population?

- Section 5 says "the CCE Gap, which is evaluated under Maximum Welfare CCE". What does it mean that it is evaluated under Maximum Welfare CCE? (What part of the definition is Maximum Welfare CCE specific?)

- First-price Sealed-bid Auction (FPSBA) is described as a multi-player game. This paper is about multi-player games. Then why is the experiment only in a 2-player FPSBA?

- Blotto is a 2-player zero-sum game. Can we compare to normal PSRO with a Nash Equilibrium meta-solver?

---

> ### Author Response · Authors · 2023-11-17
> **Response to Reviewer EpNp**
>
> We thank the reviewer for the valuable comments and are glad to discuss the suggested interesting work. We apologize for the fact that we use some notations and concepts that are implicitly defined by context, and we have elaborated on them in the paper.
>
> >The explanation of the metrics and equilibrium concept and algorithm implies that they can be used for both correlated as well as coarse correlated equilibria. However, the experiments seem to all use CCE versions and measure the CCE gap. Can the experiments be performed for CE as well?
>
> The experiments be performed for CE, but the computational cost of CE is significantly higher than that of CCE (Correlated Equilibrium with Diversity). Due to time constraints, we did not present experimental results for CE in the paper. However, we will take your suggestion seriously and supplement this part of experimental results.
>
> >Throughout the text, it's stated that we want to assign higher sampling probabilities to "more diverse policies". But what does it mean for a policy to be "more diverse" than another? Isn't diversity defined for a set of policies, not for an individual policy?
>
> Yes, diversity is usually defined for a set of policies. However, as I mentioned in the common response, some diversity measures define the difference between an individual policy and the other policies in the population. These definitions can be used to assess whether one policy is more diverse than another. Here, we want to convey that if a policy $\pi_a$ is more diverse than another policy $\pi_b$, then its difference from the entire population is larger than the difference of $\pi_b$ from the entire population.
>
> >I don't understand the (C)CE in Equation 8. When this is for CCE, what is $A_i$?
>
> For CCE, the definition of A_i is given by $A_i(\pi^′_i , \pi) = u_i(\pi^′_i , \pi_{−i}) − u_i(\pi)$, and detailed information is provided in Section 2.2.
>
> >Section 3.2 says that "calculating entropy can be challenging" -- why is that? Isn't it just $O(|N|×|\Pi|)$ where $|\Pi|$ is the size of each player's population?
>
> Sorry, we should be more rigorous here. What we want to express is that compared to Gini impurity, which is a quadratic function, the calculation of entropy involves a logarithmic function, which will become slower when frequent calculations are performed in large games.
>
> >Section 5 says "the CCE Gap, which is evaluated under Maximum Welfare CCE". What does it mean that it is evaluated under Maximum Welfare CCE? (What part of the definition is Maximum Welfare CCE specific?)
>
> The evaluation of the gap needs to be conducted under the meta-distribution, and it is unreasonable to assess it using the same distribution as itself. Therefore, we need to evaluate the gap between the DCCE we obtained and the target equilibrium under a more objective distribution. MW(C)CE is a reasonable choice because it maximizes the total payoff for all participants and is an easily computable linear formulation (Marris et al., 2021). Thank you for your suggestion, and we should include the relevant definition of MW(C)CE in the paper.
>
> >First-price Sealed-bid Auction (FPSBA) is described as a multi-player game. This paper is about multi-player games. Then why is the experiment only in a 2-player FPSBA?
>
> Thank you for your reminder. Indeed, we should provide the experimental results for 3-player FPSBA. Due to the increased complexity of 3-player FPSBA and the need for additional time resources, these results were not included in our paper. We will supplement the relevant experiments.
>
> >Blotto is a 2-player zero-sum game. Can we compare to normal PSRO with a Nash Equilibrium meta-solver?
>
> Yes, your suggestion is valid. We should indeed compare with normal PSRO using a Nash Equilibrium meta-solver. At that time, our focus was primarily on meta-solvers suitable for multiplayer games, and we overlooked the normal Nash Equilibrium meta-solver.
>
> ### Reference
> [1] Luke Marris, Paul Muller, Marc Lanctot, Karl Tuyls, and Thore Graepel. Multi-agent training beyond zero-sum with correlated equilibrium meta-solvers. In International Conference on Machine Learning, pp. 7480–7491. PMLR, 2021.

---

> > ### Comment · Reviewer_EpNp · 2023-11-17
> >
> > Thank you for your response! I will maintain my rating at this time.

---

### Official Review · Reviewer_unHf · 2023-11-02

**Soundness:** 3 good
**Presentation:** 2 fair
**Contribution:** 2 fair
**Rating:** 3
**Confidence:** 5

**Summary:**

The paper introduces a variant of the JPSRO methodology for solving meta-games but instead of a standard (C)CE equilibrium solver they use an adapted equilibrium solver that aims to find maximally diverse equilibria. The paper shows that under some tuning of internal parameters the uniqueness of the resulting solution concept follows and presents some closed formed descriptions of the solutions. Experimental results showcase that in some games the resulting solver can outperform previous solvers when the internal parameters are  "selected appropriate through extensive hyper-parameter tuning".

**Strengths:**

I think the paper examines an interesting domain of meta-game solvers and has some interesting ideas about incorporating diversity which has not been previously precisely formulated in the case of n-player games. The experimental results show some promise in terms of the value of the explored direction.

**Weaknesses:**

Although the paper studies an interesting setting the implementation feels somewhat lacking. First, I would expect the experimental results to actually showcase and discuss the actual diversity of the found solutions but this is surprisingly not really explored. Instead the paper only focuses on the extent in which these new techniques find states of low exploitability. I find this to be a critical weakness of the paper and I would be very interested in a deeper dive that tries to explore what are the properties of the solutions found and not whether they are epsilon equilibria for small enough epsilon.

Even the current experimental implementation leaves something to be desired. For example, it is stated that the stated performance of the technique is for a choice of internal parameters which is "selected appropriate through extensive hyper-parameter tuning". Is this an apple to apple comparison with other techniques?

The theoretical analysis of the paper is not particularly deep. Finally, the paper showcases some obvious typos or undefined notations that indicate a bit of rushed implementations.
Examples include:
Page 3. \Pi^{O:T}_{-i} from Marris et al paper not defined.
Page 4. In Equation 4 \rho is not defined.
Page 8 In all figures Itertions-> Iterations.

**Questions:**

Can you be a bit more precise about the hyper-parameter tuning process of $k$? For example when $k$ is very small (which you need to be small enough to guarantee uniqueness) then the solution you find is not optimizing for diversity but for maximizing the convex objective $\sigma^T \sigma$. So it is not clear to me whether the success of low exploitability is really due to optimizing for diversity.

Also in terms of an apple to apple comparisons to other techniques did you try to examine what would happen if a similar fine-tuning was explored for the other competing techniques?

---

> ### Author Response · Authors · 2023-11-17
> **Response to Reviewer unHf**
>
> Thank you for suggesting potential improvements to the presentation. We are glad that the reviewer considers the proposed method and the experiments here interesting.
>
> >Can you be a bit more precise about the hyper-parameter tuning process of $\kappa$? For example when it is very small (which you need to be small enough to guarantee uniqueness) then the solution you find is not optimizing for diversity but for maximizing the convex objective $\sigma^\top\sigma$. So it is not clear to me whether the success of low exploitability is really due to optimizing for diversity.
>
> The optimization objective of MG(C)CE is $\sigma^\top\sigma$, and it can be viewed as a scenario of D(C)CE when $\kappa$ is very small. The experimental results of MGCCE have been demonstrated in Figure 1, providing a comparison with the results of DCCE.
>
> >Also in terms of an apple to apple comparisons to other techniques did you try to examine what would happen if a similar fine-tuning was explored for the other competing techniques?
>
> To the best of our knowledge, there is currently no technology similar to ours. That is, no one has employed the diversity measure to define an equilibrium in a manner similar to our approach. The baselines we use also do not involve such operations. Therefore, it is challenging to conduct similar fine-tuning for other technologies at this point.

---

> > ### Comment · Reviewer_unHf · 2023-11-21
> > **Response**
> >
> > I thank the authors for their response. Although I find the paper interesting, I believe that my original criticisms still apply and hence I do not believe that in its present from the paper suffices for an ICLR publication.

---

### Author Response · Authors · 2023-11-17
**Common Response [1/2]**

We gratefully acknowledge all reviewers for their insightful comments and suggestions. Three critical issues summarised from the comments and suggestions in the common response will be discussed as follows. The remaining issues will be addressed separately.
### 1. Motivation for Diverse Correlated Equilibrium
Diversity measures the variety of policies. Specifically, existing diversity is measured either by the divergence of the occupancy measure corresponding to the policies of different agents, or by the distance of vectors composed of the payoffs obtained by the agents facing different opponents. For the former, such measure is usually defined as the minimum or average divergence between the occupancy measure of each policy and the occupancy measures of other policies in the population; while for the latter, diversity is either defined as the norm of the difference of the payoff vector of the current policy and the payoff vector of other policies, or as a function of the norm of the payoff matrix.

Existing diversity-aware PSRO algorithms usually use diversity as the regularization term of the initial optimization goal (ie, expected return). In the Best Response stage of the algorithm, the entire optimization objective is solved to obtain new policies. Our perspective is focused on the steps following the Best Response phase, at which point a new policy has just been solved and added to the population. We can still calculate the diversity of this population, especially as done by Convex Hull Enlargement, Occupancy Measure Mismatching, and Policy Space Diversity, to calculate the distance (divergence or norm) between each policy and other policies in the population. From this perspective, these diversities, or distances, are measures of how similar different policies within a population are to each other, and can reflect which policies are farther or closer to the whole population.

On the other hand, the necessity of diversity is manifested in training adaptive policies. Specifically, diverse training opponents help gradually eliminate the weaknesses of policies, allowing policies to adapt to a wide range of opponents (Liu et al., 2021). Interaction is sparse. In different PSRO algorithms, the meta-strategy of opponents determines the probability of each policy in the opponent population being sampled. Therefore, a reasonable approach is to provide greater sampling opportunities for opponent policies that are farther from the population (i.e., more diverse), which can avoid repeatedly facing similar opponent policies in different training rounds, thereby enhancing the adaptability of policies more quickly. However, currently, no work has considered this aspect. Our work addresses this in the Meta Solver phase by incorporating the diversities calculated in the Best Response phase into the computation of the meta-strategy. Our optimization objective is $\max_{\sigma} (\textbf{f}^\top \sigma)^2$, which is a variation of the original objective $\max_{\sigma} \textbf{f}^\top \sigma$ (where both $\textbf{f}$ and $\sigma$ are non-negative). This objective function is essentially a weighted average of diversities, with the weights being the meta-strategy. Given the diversities, maximizing this objective function means assigning higher weights to larger diversity terms, i.e., providing greater sampling opportunities for more diverse policies.

---

> ### Author Response · Authors · 2023-11-17
> **Common Response [2/2]**
>
> ### 2. Advantages of Diverse Correlated Equilibrium
> 1. It enhances the performance of the PSRO framework in multiplayer games, particularly effectively reducing the gap between the solved meta-strategy and the target equilibrium.
> 2. It introduces diversity into the equilibrium-solving process. Diversity, as additional supplementary information, provides a positive gain for equilibrium convergence: opponent policies with greater diversity have a higher chance of being sampled and used to train the current policy, thereby improving the adaptability of the current policy.
> ### 3. Convergence of DCO
> As far as our knowledge extends, PSRO-based algorithms, or extensions of PSRO in multiplayer games like JPSRO-based algorithms, can only be proven for convergence and not for deriving regret bounds or exploitability bounds. The D(C)CE we propose can be considered a type of meta-solver, and DCO is a JPSRO-based algorithm that employs D(C)CE as its meta-solver. Therefore, concerning the convergence of the DCO algorithm, we can directly refer to the conclusions in (Marris et al., 2021):
>
> *\'When using a (C)CE meta-solver and (C)CE best response in JPSRO((C)CE) the mixed joint policy converges to a (C)CE under the meta-solver distribution.\'*
> ### Reference
> [1] Xiangyu Liu, Hangtian Jia, Ying Wen, Yujing Hu, Yingfeng Chen, Changjie Fan, Zhipeng Hu, and Yaodong Yang. Towards unifying behavioral and response diversity for open-ended learning in zero-sum games. Advances in Neural Information Processing Systems, 34:941–952, 2021.
>
> [2] Luke Marris, Paul Muller, Marc Lanctot, Karl Tuyls, and Thore Graepel. Multi-agent training beyond zero-sum with correlated equilibrium meta-solvers. In International Conference on Machine Learning, pp. 7480–7491. PMLR, 2021.

---

### Meta-Review · Area_Chair_Ge3B · 2023-12-07

**Metareview:**

a) Claims: The paper proposes a variant of JPSRO that solves the meta-game by finding a maximally diverse CE/CCE, gives an analytical form for its solution, and empirically evaluates its performance.

b) Strengths: The reviewers all agreed that the idea of taking account of diversity in meta-game solving is an interesting direction to explore.

c) Weaknesses: The reviewers were unanimous that this paper did not meet the bar for clarity of expression.  Most reviewers found the empirical evaluation unconvincing; the proposed method did not convincingly outperform its competitors, and the proposed method appears to have required intensive hyperparameter tuning.  Many reviewers also found the overall motivation unclear (i.e., why should we want more diverse CCEs in the first place?  Empirical performance could have been a good justification if the performance had been better, but it wasn't.)  One reviewer noted that if we should care about solution diversity for its own sake, it's strange that the diversity of the solutions was never directly evaluated.

**Justification For Why Not Higher Score:**

The reviewers were unanimous that this paper does not meet the bar, due to issues with clarity and significance of the contribution.

**Justification For Why Not Lower Score:**

n/a

---

### Decision · Program_Chairs · 2024-01-16

Reject